# Process evaluation of co-designed interventions to improve communication of positive newborn bloodspot screening results

Jane Chudleigh ![ORCID],[1] Pru Holder,[1] Louise Moody,[2] Alan Simpson,[3,4] Kevin Southern,[5] Stephen Morris ![ORCID],[6] Francesco Fusco,[7] Fiona Ulph,[8] Mandy Bryon,[9] James R Bonham,[10] Ellinor Olander[1]

For numbered affiliations see end of article.

**Correspondence to**
Dr Jane Chudleigh;
j.chudleigh@city.ac.uk

## ABSTRACT

**Objective**  To implement and evaluate co-designed interventions to improve communication of positive newborn bloodspot screening results and make recommendations for future research and practice.

**Design**  A process evaluation underpinned by Normalisation Process Theory.

**Setting**  Three National Health Service provider organisations in England.

**Participants**  Twenty-four healthcare professionals (7 newborn screening laboratory staff and 24 clinicians) and 18 parents were interviewed.

**Interventions**  Three co-designed interventions were implemented in practice: standardised laboratory proformas, communication checklists and an email/letter template.

**Primary outcome measures**  Acceptability and feasibility of the co-designed interventions.

**Results**  Auditing the implementation of these interventions revealed between 58%–76% of the items on the laboratory proforma and 43%–80% of items on the communication checklists were completed. Interviews with healthcare professionals who had used the interventions in practice provided positive feedback in relation to the purpose of the interventions and the ease of completion both of which were viewed as enhancing communication of positive newborn bloodspot screening results. Interviews with parents highlighted the perceived benefit of the co-designed interventions in terms of consistency, pacing and tailoring of information as well as providing reliable information to families following communication of the positive newborn bloodspot screening result. The process evaluation illuminated organisational and contextual barriers during implementation of the co-designed interventions in practice.

**Conclusion**  Variations in communication practices for positive newborn bloodspot screening results continue to exist. The co-designed interventions could help to standardise communication of positive newborn screening results from laboratories to clinicians and from clinicians to parents which in turn could improve parents' experience of receiving a positive newborn bloodspot screening result. Implementation highlighted some organisational and contextual barriers to effective adoption of the co-designed interventions in practice.

## STRENGTHS AND LIMITATIONS OF THIS STUDY

⇒ This is the first known study that has used co-designed interventions to improve communication of positive newborn bloodspot screening results.

⇒ Healthcare professionals involved in the present study were employed in three different National Health Service organisations, increasing transferability of the findings.

⇒ Healthcare professionals were recruited via email; those with a pre-existing interest in this topic may have been more likely to self-select into the study. These may also communicate results differently than providers who did not participate in the study.

⇒ The study included healthcare professionals involved in managing, and parents of children diagnosed with, one of the nine conditions currently included in the newborn bloodspot screening programme in England; previous work has mainly focused on cystic fibrosis and sickle cell disease.

⇒ COVID-19 hindered implementation of the co-designed interventions and related data collection.

**Trial registration number** ISRCTN15330120.

## INTRODUCTION

Each year in England, almost 10 000 parents are informed of their child's positive newborn bloodspot screening (NBS) result around 2–8 weeks, depending on the condition, after birth.[1 2] NBS currently includes nine conditions; sickle cell disease (SCD); cystic fibrosis (CF); congenital hypothyroidism (CHT); phenylketonuria (PKU); medium-chain acyl-CoA dehydrogenase deficiency (MCADD); maple syrup urine disease (MSUD); isovaleric acidaemia (IVA); glutaric aciduria type 1 (GA1); and homocystinuria (HCU) (pyridoxine unresponsive) – the latter six collectively referred to as inherited metabolic diseases (IMDs).



Communicating positive NBS results is not an event but a process that starts from the moment the result is identified by the NBS laboratory (NBSL) as being above the agreed analytical 'cut-off' and ends when the parents are given the definitive diagnosis for their child.[3] The clinical spectrum in screen positive cases varies enormously and consequently the message to parents needs to be carefully crafted to prepare for a range of outcomes; communication of positive NBS results is a subtle and skilful task which demands thought, preparation and evidence to minimise potentially harmful negative sequelae.[4–8]

Guidance regarding the content and best mode of communication between healthcare professionals (HCPs) and parents is variable for the different screened conditions[9 10] and is often not evidence-based. Studies have shown that receiving a positive NBS can make parents feel anxious and stressed[6 11–14] and that the knowledge of the HCP and the approach used can help alleviate this[13–15] Parents who were dissatisfied with the initial information often complained about a lack of explanation for the test result or that information was superficial.[12] It is clear that parental information needs are variable and that improvements are needed.[11 13]

Poor, or inappropriate, communication strategies for positive NBS results can influence parental outcomes in the short term[4–7 11–13] but may also have a longer-term impact on children and families.[8 16] Therefore, exploring strategies that can alleviate distress associated with the initial communication of a positive NBS result is crucial to improving parents' experiences.

Existing evidence supports the importance of ensuring the initial communication of positive NBS results is handled sensitively, considers individual parent characteristics, to minimise parental distress and consequences of this distress as well as the knowledge and experience of the person imparting the result. However, to date, while studies have explored experiences of receiving positive NBS result and strategies for improving communication,[6 7 17 18] no studies have focused on designing implementing or evaluating such strategies in practice. The purpose of the current study was to implement and evaluate three interventions to improve communication of positive newborn bloodspot screening results and make recommendations for future practice and research. The interventions were co-designed by parents and HCP; this is described elsewhere.[19]

## METHODS

Experience-based co-design[20] was used to develop co-designed interventions to improve communication of positive NBS result to families.[21] A process evaluation underpinned by Normalisation Process Theory (NPT)[22 23] was used to study the implementation and assimilation of the co-designed interventions into routine practice as part of a larger programme of work.[3 21 24] These included: a standardised laboratory proforma for communication of positive NBS results from the NBSLs to clinicians;

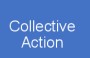

**Coherence**
- Are the interventions easy to describe?
- Are they distinct from other interventions?
- Do the interventions have a clear purpose?
- Does it fit in with the overall goals of the organisation?

**Cognitive Participation**
- Is it possible to recruit the staff from each study site? If <50% of staff approached, agree to participate, consider stopping in consultation with PPI group.
- Are staff willing to invest the time required to implement the interventions into practice? If drop out rate ≥50% then consider stopping in consultation with PPI group.

**Collective Action**
- Is the training required too time consuming to make this feasible in practise?
- Are the interventions compatible with existing resources?

**Reflexive Monitoring**
- Is implementation of the intervention sustainable?
- Does the qualitative data imply any negative psychological sequelae from the implementation of the interventions? Any 'incidents' should be reported to and discussed with PPI group.
- Are the interventions being implemented as planned (fidelity)? If not are the adaptations appropriate for local context?

**Figure 1** Success criteria for implementing the co-designed interventions.[22] PPI, patient and public involvement.

standardised communication checklists for communication of positive NBS results to the family; and a letter/email template for providing information to the family immediately after communication of the initial NBS result. Staff in each study site were able to choose from a variety of training options including face-to-face (in person or remotely) individual or group training (this was condition specific to ensure the correct interventions were presented to the relevant staff) via narrated PowerPoint presentations and/or annotated PowerPoint presentations. These training materials were also made available on the study blog. Success criteria (figure 1) were defined to ensure that implementation of the co-designed interventions was acceptable and feasible.

### Patient and public involvement
Patient and public involvement (PPI) was fundamental to the design and conduct of this study. Eight parents of babies who had received a positive NBS screening result for one of the nine screened conditions formed a PPI advisory group who met prior to, during and following data collection. Their suggestions were incorporated into the study design, the data collection tools and the data analysis and presentation. The PPI group were presented with data from the annual reports of the NBS programmes and made suggestions as to which sites should be used for the implementation phase of the work. Initial findings were presented to members of the PPI group during regular 6 monthly meetings. In addition, we obtained the views of representatives from charities for the screened conditions including Metabolic Support UK, the British Thyroid Foundation, the CF Trust and the Sickle Cell Society.

### Setting
The interventions were introduced into three National Health Service (NHS) provider organisations in England (the study sites) served by two NBSLs that had been involved in the co-design process.[19]

### Inclusion and exclusion criteria

In order to explore the actual and perceived usefulness of the co-designed interventions, parents who had been given their child's positive NBS result during the previous 12 months for whom the co-designed interventions had and had not been used were included. This enabled us to explore if the perceived perspectives and actual experiences of the interventions were comparable.

NBSL staff involved in processing NBS samples and clinicians including consultants and nurses specialists involved in communicating positive NBS results in the previous 6 months, were invited to take part in the study. HCPs (NBSL staff and clinicians) who had not been involved in NBS in the last 6 months or who had personal experience of receiving a positive NBS result were excluded.

### Recruitment and sampling

HCPs were sampled purposively based on their experience with the phenomena of interest, that is, they had used the co-designed interventions in practice. HCPs provided all parents who had experienced the interventions with an information sheet about the study and asked if they could share their details with the research team. Once the details were received, parents were contacted by the research team and invited to participate in the study. In addition, a purposeful sample of parents who had received a positive NBS result without the co-designed interventions were also identified by HCPs. Written informed consent was obtained from all participants.

### Data collection

Following training with HCPs in each of the study sites, the co-designed interventions were implemented in March 2020 and then following a pause due to COVID-19, from September to December 2020.

### Audit of completion of co-designed interventions

The fidelity of the co-designed interventions was assessed in the three NHS Trusts. Staff were asked to send the research team all completed laboratory proformas and communication checklists so these could be audited in terms of accuracy and completeness. These were redacted to maintain patient confidentiality.

### Semi-structured interviews

HCPs were invited to take part in semi-structured interviews to ascertain their views regarding the acceptability, feasibility and usability of the co-designed interventions following implementation. The interview questions (online supplemental file A) were based on those proposed by the developers of the NPT approach[22 22 23 23] and the success criteria (figure 1). The purpose was to explore the views of the interventions and perceptions of factors that were influential (mechanisms of impact and context).[25]

Parents who had received a positive NBS screening result both with and without the co-designed interventions were also invited to take part in semi-structured interviews to ascertain their experiences and perceptions of the co-designed interventions, respectively (online supplemental file B).

### Data analysis

#### Audit of completion of co-designed interventions

Accuracy and completeness of the co-designed interventions was audited; percentages were calculated for each item.

#### Semi-structured interviews

All interviews were audio-recorded and transcribed verbatim. Interviews undertaken with HCPs were subject to framework analysis.[26] Success criteria (figure 1) were developed using NPT[22 23] and provided the framework for analysis of these data. In the first stage (familiarisation), two members of the research team (JC and PH) familiarised themselves with the data by reading the interview transcripts. In stage 2 (developing a theoretical framework), key, recurring themes in the same interview transcript were compared with the a priori success criteria (figure 1) by the same two members of the research team. In stage 3 (indexing) two members of the research team (JC and PH) coded data from a further two interviews while identifying relevant participant quotes for the identified themes/subthemes from each interview. These were compared and 90% intercoder reliability was achieved. In stage 4 (charting) the same two members of the research team agreed on a final framework with four subthemes and data were summarised in a thematic chart in an Excel spreadsheet. In the final stage (synthesising) the two members of the research team created a summary of the main descriptive comments and developed an explanatory account.

Interviews with parents were analysed thematically; an inductive approach to data analysis was used and themes generated using a latent approach to provide a deeper understanding of opinions regarding the proposed interventions.[27] Two members of the research team (JC and PH) coded one interview transcript separately. These codes were then compared with inform and align code development[28] and a code book was developed.[29] A further two transcripts were then coded separately by the same two members of the research team using the code book. These separately coded transcripts were then compared; intercoder reliability was 87%. Following this, the remainder of the transcripts were divided between the same two members of the research team. This was an ongoing, iterative process; new codes were developed and the definition of codes refined as analysis progressed. Any proposed changes to the code book were discussed where relevant after each interview transcript had been analysed to ensure consensus; the final version of the code book (V.3) was developed after a further three transcripts had been analysed. Once coding had been completed, all data for each code were compared with ensure consistency in coding.

### Positionality and reflexivity

Members of the study team (JC, JRB, LM, FU, MB and KS) have been involved in or continue to undertake a variety of roles and activities associated with the NBS programme in the UK. It is acknowledged that this could have led to potential bias during data collection and analysis. However, this was balanced by other members of the research team who had previously had minimal involvement in NBS (EO, AS, SM, FF and PH). Data collection and analysis was mainly undertaken by JC and PH who fall within both camps. Neither JC nor PH were employed in the organisations where data collection was undertaken.

## RESULTS

### Training

Training for the implementation of the co-designed intervention was undertaken with 23 staff in study site 1, 9 staff in study site 2 and 14 staff in study site 3. Forty-one staff were trained during face-to-face sessions and five were trained online. Staff were asked to provide feedback at the end of the training sessions using a 5-point Likert scale consisting of statements ranked from strongly agree to strongly disagree. Twenty-nine staff (63%) provided feedback. These were scored numerically so that strongly agree was awarded a score of 5 and strongly disagree was awarded a score of 1; a score of 3 would therefore have indicated a neutral response, a score over 3 would indicate positive feedback and under 3 would indicate negative feedback. Responses ranged from 4.3 to 4.6 (median 4.5).

### Staff interviews

Thirty-one HCPs were approached across the three study sites (7 NBSL staff and 24 clinicians); 24 were interviewed (median 27.1, range 10.59–57.2 min). Seven were interviewed from Site 1 (four did not respond to the invitations); nine were interviewed from Site 2 (all responded to invitations) and eight were interviewed from Site 3 (three did not respond to invitations). This is summarised in box 1.

### Interviews with parents

Twelve parents (seven from Site 1, four from Site 2 and one from Site 3) who had received a positive NBS result but who had not experienced the interventions were

| Box 1 | Staff interviews |
|---|---|

**Site 1:** *n=7*
1 NBSL staff, 6 clinicians (2×IMD, 2×SCD, 2×CF)
**Site 2:** *n=9*
3 NBSL staff, 6 clinicians (2×IMD, 4×CHT)
**Site 3:** *n=8*
1 NBSL staff, 7 clinicians (3×SCD, 4×CF)

CF, cystic fibrosis; CHT, congenital hypothyroidism; IMD, inherited metabolic disease; NBSL, newborn bloodspot screening laboratory; SCD, sickle cell disease.

**Table 1** Interviews with parents

| Parents who had received: | Site and condition | |
|---|---|---|
| A positive NBS result but had not experienced the interventions, n=12 (8 mothers, 4 fathers) | **Site 1** | |
| | CF | 2 |
| | MCADD | 3 |
| | PKU | 2 |
| | **Site 2** | |
| | MCADD | 2 |
| | IVA | 2 |
| | **Site 3** | |
| | CHT | 1 |
| A positive NBS result and had experienced the interventions, n=6 (3 mothers, 3 fathers) | **Site 1** | |
| | PKU | 2 |
| | **Site 3** | |
| | CF | 4 |

CF, cystic fibrosis; CHT, congenital hypothyroidism; IVA, isovaleric acidaemia; MCADD, medium-chain acyl-CoA dehydrogenase deficiency; NBS, newborn bloodspot screening; PKU, phenylketonuria.

interviewed (median 29.4, range 16.5–36.4 min), none of those approached declined. Eight parents who had received a positive NBS result and had experienced the interventions were approached, six were interviewed (median 28.5, range 15.4–43.3 min). Two parents from study Site 1 did not respond to arranged telephone calls and consequently were not interviewed. Those interviewed consisted of two parents from Site 1 who had a child with PKU and four from Site 3 who had children with CF. This is summarised in table 1. Interview responses from parents who had experienced the interventions are denoted by ($^+$), and parents who had not experienced the interventions by ($^-$).

### Intervention fidelity

Two NHS Trusts (Sites 2 and 3) were served by one NBSL which had implemented the co-designed laboratory proformas. The other, Site 1 was served by the other NBSL and had only implemented the proforma for CF and had therefore only completed the form on one occasion. Feedback was therefore sought from staff in all three Trusts but for the latter, this focused on exploring any challenges they had experienced which had led to them not fully implementing the co-designed interventions in order to determine potential barriers.

Two clinical teams in Site 1 (metabolic and CHT) and one clinical team in Site 3 did not implement the co-designed interventions; feedback was still sought from these to determine any barriers to implementation; one team in Site 1 did not respond.

Seventy completed laboratory proformas and 16 communication checklists were provided by the study sites. The NBSL that served Sites 2 and 3 chose to adopt the proforma from March to December 2020, and were

**Table 2** Audit of completed study laboratory proformas and communication checklists

| Study laboratory proforma | | | |
|---|---|---|---|
| | Number sent from each site | Average (range) % complete | |
| Condition | | Side 1 | Side 2 |
| **Site 1** | | | |
| CF | 1 | 69 (N/A) | 0 |
| **Site 2** | | | |
| CF | 18 | 69 (62–77) | 0 |
| CHT | 36 | 75 (61–82) | 0 |
| GA1 | 1 | 67 (N/A) | 0 |
| IVA | 1 | 76 (N/A) | 0 |
| MCADD | 2 | 68 (64–73) | 0 |
| PKU | 8 | 62 (56–84) | 0 |
| **Site 3** | | | |
| SCD | 3 | 58 (54–58) | 19 |
| **Checklists for initial communication of positive NBS result** | | | |
| Condition | No. completed | Average % complete | |
| **Site 1** | | | |
| MCADD | 1 | 80 (N/A) | |
| PKU | 1 | 68 (N/A) | |
| SCD | 1 | 76 (N/A) | |
| **Site 2** | | | |
| CHT | 2 | 72 (66–78) | |
| PKU | 4 | 43 (32–68) | |
| **Site 3** | | | |
| CF | 7 | 57 (35–87) | |

CF, cystic fibrosis; CHT, congenital hypothyroidism; GA1, glutaric aciduria type 1; IVA, isovaleric acidaemia; MCADD, medium-chain acyl-CoA dehydrogenase deficiency; NBS, newborn bloodspot screening; PKU, phenylketonuria; SCD, sickle cell disease.

therefore able to provide completed proformas for a 9-month period. The other NBSL and clinicians in all sites, only collected data using the proforma and checklists during March 2020 and then September–December 2020; this is reflected in the number of proformas and checklists returned for each site in table 2.

The NBSLs completed between 58%–76% of the items on the first page of the laboratory proforma. The most common items not completed on the laboratory proforma included: the case/laboratory ID; the designation of the person making the referral; legacy names (CF); haemoglobin S (HbS) mutations (SCD); General Practitioner (GP) phone number; results of other conditions included in NBS; and feedback from the clinical team. The second page, which contained further information about the NBS test results and a checklist regarding completion of the referral process was not completed indicting a lack of fidelity for this section of the intervention. Clinicians indicated that reducing the length of the laboratory proforma so it fitted on one side of paper for ease of completion would be preferable.

NBSL, Site 2, Participant 1: It's good to try and keep them on one A4 piece of paper if possible…case laboratory ID stuff is on the second page. We don't need that…we can get that from our [IT system].

The original objective of the study was to produce a checklist for communication of the initial positive NBS result. However, during the co-design phase, clinicians in the study sites and parents indicated they would like a series of checklists that catered for all communication around the positive NBS result. Therefore, separate checklists were developed for the initial communication, the initial clinical visit and subsequent clinical visits. However, most clinicians consistently chose to only use the checklists for the initial communication. Therefore, data in table 2 is only presented for this section of the co-designed intervention. Completion ranged from 43%–80%. The most common items not completed on the initial communication checklist included: diagnosis date/age; weight; gene mutations (SCD); GP details; health visitor details; summary information from the first contact. Therefore, while in theory staff favoured the idea of having more information in one place about the child and family, this was not borne out in practice. This was supported by the interview data; some staff questioned the usefulness of the checklists for the first and subsequent clinical visits.

Clinician 1, Site 2: I like the initial consultation bit, or the first visit bit. I like that bit but I have to say I don't really use much of the rest of the pages.

However, other clinicians indicated that having all of the information together regarding communication with parents following a positive NBS result was useful.

Clinician 1, Site 3: I think it's useful for the clinician to know what they've said last time and build on that information the next time they see them and then it's useful when they end up seeing someone completely different to be able to look back and just remind parents it's already been covered but let's go through it again. You know, I think it's quite a good prompt for people to be systematic about the information they give out.

### Implementation of the interventions

Three themes were identified in relation to the implementation of the laboratory proforma and the communication checklists: Coherence of the intervention; compatibility and resource use; and ease of completion and layout. The first two themes were also applicable in relation to the email/letter to parents.

### Laboratory proforma
#### Coherence of the intervention

Staff in the NBSL that served Sites 2 and 3 and who had fully implemented the interventions felt that the project intentions were positive and they improved current processes. Consequently, they were keen to implement

the laboratory proformas during the study period and keep using the proformas once the study had ended.

In addition, feedback from this NBSL (Sites 2 and 3) suggested that they included more information than the previous forms they had been using which was viewed favourably.

> NBSL, Site 2, Participant 3: I think they're better from the point of view that it tries to capture more information than we used to have. Especially for the carriers and stuff…I can see the difference when I go to fill in one of the carrier ones or one of the old expired ones.

Furthermore, in relation to CHT, staff in this NBSL felt that the study proformas would help clinicians who were not based in tertiary centres to understand the next steps in the process for these babies. This also meant the NBSL staff felt more confident that the babies would be followed up appropriately.

> NBSL, Site 2, Participant 1: My concern always, again, going back to CHT, is that the person who we give the result to may not always have the knowledge available of what to do next. Having that checklist for them about what they're meant to do…It's useful for them, especially if their consultant's not present…we didn't really put a checklist in our referrals before for CHT…that checklist will help most locum paediatric registrars who we refer the result to, to give them some guidance what to do next.

Staff in the same NBSL felt that having standardised national laboratory proformas would be useful if NBSL staff or clinicians moved from one site to another as they would already be familiar with the processes used.

> NBSL, Site 2, Participant 2: I think that would be really advantageous. I think it's meant to be a national screening programme and we are meant to do the same thing, we're meant to follow the same protocols. And if we use the same paperwork, we would make our lives easier, we would make people moving between labs, people moving between clinical teams, any of the above in the screening pathway, would simplify it for them if everybody was doing the same thing.

The other NBSL (Site 1) was not aware that other laboratories were using different forms to refer positive NBS results to clinical teams and therefore could not see the purpose of the intervention.

> NBSL, Site 1, Participant 1: I naively assumed that every lab was using the national templates that have been recommended and are in all the lab guides. So, I don't know why this has been reinvented, really.

This site (Site 1) had only implemented the laboratory proformas for one condition and stated that as they were very similar to their existing processes, they did not understand the rationale for the proposed changes and felt they would take too long to implement.

### Compatibility and resource use

Difficulties in terms of operationalising standardised processes were raised due to individual NBSLs following different processes:

> NBSL, Site 2, Participant 3: Each lab, even though we follow the same standards, the same guidelines, there's always a different way of doing things, and so what's inconsequential to one lab may seem consequential, but then my take on this is if you've got eleven labs doing it one way and the twelfth lab is doing it a completely different way then, the other lab should in theory get in line, because eleven labs are doing it all the same way.

This was linked to similar concerns regarding how the laboratory proformas would fit into the different computer systems that were being used nationally by different NBSLs. However, a proposed solution involved changing processes so the standard proformas get automatically populated form the existing computer system to avoid transcription errors.

For Sites 2 and 3 (served by one NBSL) that had fully implemented the laboratory proformas, it was acknowledged that using a different form was initially more time consuming which could act as a barrier to implementation. However, feedback suggested once familiar with the layout of the form, this resolved quite quickly:

> NBSL, Site 2, Participant 3: At the beginning you may find it a little bit, I don't know, scary, or it will take more time but after doing quite a lot it's alright.

Also, that this was offset by the fact that the study proformas allowed all the required information to be collated in one place which had the potential to save time in the long run.

> NBSL, Site 2, Participant 3: As a lab person, it makes me having sort of a lot of information of the day together so I don't have to look here and there. Like let's look at the card, let's look at the referral letter, let's look at this and that. And I think the clinicians should be happier because they have them all together. Definitely in terms of the lab, I felt that it's much more informative than the ones we had.

This view was shared by clinicians who had received the laboratory proformas who fed back that the laboratory proformas had been time saving and helpful in terms of sharing information and reducing the number of resources they needed to consult to gather information about the family.

> Clinician 1, Site 3: We're not opening lots of attachments to find the information we need, it's all there in one place which is good…I quite like this actually. I think it's a big improvement on the old one.

### Ease of completion and layout

Feedback suggested that the layout of the laboratory proformas made them easy to use in practice. However, there were some boxes that the NBSL identified were labelled slightly differently in terms of terminology than the laboratories were used to and therefore it was recommended that these could be changed. Furthermore, one of the sites also raised queries about who was responsible for completing certain parts of the laboratory proforma and thought this could be made more explicit. Consequently, study sites also made recommendations for minor changes to the proformas such as combining sections to avoid duplication.

> NBSL, Site 3, Participant 1: It's, kind of, knowing what you want filled in when by which person, which lab or other parties, so that whoever's using the form is aware which ones they're filling in.

## Communication checklists
### Coherence of the intervention

Clinicians viewed the purpose of the communication checklist as multifaceted. Similar to the laboratory proformas, many clinicians indicated that standardising communication nationally would be advantageous in terms of improving communication with families of children with positive NBS results.

> Clinician 1, Site 1: For us, this is our patient's journey. We want the best experience for them and we want to make sure that every family have the same experience and, you know, the same information. So, for us to standardise something, I think, is crucial.

This view was also shared by parents who had not experienced the interventions. These parents were aware that practices differed throughout the country with regard to how positive NBS results were communicated to parents and were keen for this to be remedied:

> Site 1, Mother, MCADD⁻: More consistent across the UK. That would be great, wouldn't it? That everybody has the same consistent, great support right from the beginning.

Parents who had experienced the interventions also thought the checklists were reassuring.

> Site 1, Father, PKU⁺: Yes, checklists are good aren't they. I do like a checklist to be honest. But, yes, they are very easy, simple. …a checklist is easier, you see it in full, it's explanatory and it's reassuring.

Parents who had not experienced the interventions also stated they would reduce inconsistency and ensure parity in terms of the experience parents had of receiving their child's positive NBS result.

> Site 1, Father, PKU⁻: A checklist is a checklist regardless of what industry, what religion, what education you are isn't it…so I don't see any reason why, it's not

like you're asking someone to log on to a computer and talking a different language or, you know, work out some kind of maths equation, it's simple, standard, uniform process across the world, a checklist.

It was also felt that using the checklists could be helpful in terms of facilitating communication both within and across clinical teams. The former referred to the family seeing different team members during clinic visits.

> Clinician 2, Site 3: I think it's a useful thing because we try to stick to the same consultant for the first few visits at least, so there's some continuity. But that can't always be guaranteed, if you're going on holiday…it's really helpful to have the tick box to say that this is covered, that's covered, because there are gaps of things that have not been covered, then you know that you can address them, you know, when you see them. I think it is helpful, yes.

Others clinicians felt that standardisation was not always possible or desirable due to the need to accommodate parental reactions and tailor information accordingly; thus, highlighting the perceived importance of the skills and attributes of the person communicating the result. However, it was acknowledged that it was still important that certain pieces of information were communicated to families and the communication checklist could help with that.

> Clinician 3, Site 2: I think it's too nuanced and complicated and different families need different things. I don't think it's the sort of thing you can really standardise, whereas our initial contact is, by its very nature, pretty structured. You've got a very small amount of time and it's just a phone call, and you're focussed on the really important bits of information to get the family in and give them the basic kind of information.

Nevertheless, the same clinician thought the checklists would be useful for new members of staff, nurses and doctors, who may be less familiar with processes and procedures. Other benefits included acting as an aide memoir and improving accountability.

> Clinician 1, Site 1: We see families and we have lots of conversations, but we don't actually always document what we talk about. And actually, from an accountability point, we probably should. So, actually it'll probably make us work better.

Furthermore, parents who had experienced the communication checklist indicated that separating information in terms of what needed to be covered in the initial communication, the first clinical appointment and subsequent visits would also be preferable to avoid overloading them with information.

> Site 1, Mother, PKU⁺: I think what would have helped me is just have it in more of a section of, right, from nought to six months this is what you need to focus on…And then from two years on, whatever the

brackets are. Just because, we were asking questions about how it's going to affect [baby] when she's twenty. And on that day, at that moment, you probably do worry about all that stuff, but it needed just to be said, 'Right, calm down, let's just focus on the next six months, and this is what it's going to look like'.

Parents who had not experienced the interventions shared similar feelings and felt that using a checklist could be beneficial from a HCP perspective to help with pacing and tailoring information to families. For some, this meant that it would avoid repetition.

Site 1, Father, MCADD⁻: At least then they're not repeating themselves to us, and we're not asking the same questions, they're not asking the same questions of us.

For others, this meant that concepts that they had found difficult could be revisited:

Site 1, Father, PKU⁻: In the early days of having it, repetition is kind of healthy, it reinforces our knowledge of it, like I say my memory is terrible so, you know, hearing the doctors saying it to me a few times is pretty handy.

### Compatibility and resource use

Clinicians who had not used the communication checklists indicated that other potential barriers to using them included personal preference, clinical experience and knowing what to tell parents already. One participant in particular called the checklist 'a bit redundant and… moderately insulting' Clinician 3, Site 3. This manifested in a reluctance to move away from a system they were comfortable and familiar with.

Clinician 3, Site 1: It feels like a tick box exercise for me. So that's my challenge, is that, and I don't use them in a way that probably somebody that was new would use them because they would probably be more rigid at going through and making sure they had understood all those things. Whereas I, sort of, already know the key things that I have got to cover and I do cover them. So, it's more paperwork for me to do/fill.

In terms of time needed to implement the interventions, opinions of clinicians were divided. Those who had not implemented the interventions anticipated them taking more time to complete than current processes. However, those who had used the co-designed interventions in practice indicated they had taken less time than their current processes.

Clinician 4, Site 3: I think it [communication checklist] just makes it a bit more streamlined. It's definitely not made it longer …these ones have been quicker.

Reasons for not using the interventions were also explored which highlighted the importance of perceived

gatekeepers buying into the proposed changes and their influence on others even if they did not share the same view. This included staff instructing others not to use the checklists as they were seen as a 'tick box exercise'.

Clinician 1, Site 1: The initial push-back from [member of staff] was that it's a checklist and we see them in the person, and we don't have the paperwork out and go, 'Tick, tick, tick.' But actually, you're sitting at a computer now and not doing home visits. So, you should be using this. You've got no excuse.

Concern was also raised with regard to what would happen with the checklists following communication of the positive NBS result and who the information would be shared with.

Clinician 3, Site 3: [The proforma and checklists] contain a lot of patient information and addresses and dates of birth, and it's all personal information that has to be stored properly and securely, and you have to, I guess, think what the form's for and what are you going to do with it?… It's not clear who would want to see it in the future, where you would file it, what it's for, who it would help.

Other clinicians had already considered this issue and stated it would be useful to have the checklist at the front of the child's medical notes which would be stored according to hospital policy and therefore be compliant with General Data Protection Regulation (GDPR) requirements.

### Ease of completion and layout

Clinicians expressed mixed feelings about completing the checklists; these were generally divided into those who had used them and those who had not. Some clinicians who had implemented the communication checklists had experimented with how best to use them in practice.

Clinician 4, Site 2: I think what I did the very first time I used it, I did my spiel and then I went through and said, 'Oh, actually, did I mention that-,' I think there were, like, a couple of things that I hadn't said, maybe, 'Have you got a question that you want to write down?' or something like that. I might not have thought to offer them. So, it was useful to have. And then, I think, the second time I did it I just followed your pro forma specifically as opposed to doing my usual one, and then, I think, I maybe just expanded bits that I wanted to.

Others indicated that they had used the checklists from the outset and had found them logically presented and a useful prompt for the topics that they needed to cover during their conversations with parents. This view was shared by parents who had experienced the communication checklists. Parents indicated that they liked the succinct, straightforward nature of this when it had been used to inform them of their child's positive NBS result.

Site 3, Mother, CF[+]: I got run through they received the results, obviously, to the screening test and said that one of the ones that came out was the CF gene …they said that the CF meant that they wanted me to come to hospital the next day and that was basically it, really. I think it was, sort of, good to be just blunt to the point of 'Yes, she has it,' and definitely to come in the next day.

Parents who had not experienced the interventions also indicated it was useful to have 'essential information' and 'optional additional information' separated on the initial communication checklist. They felt this would facilitate tailoring of information provision during this first contact to accommodate parental reactions and how receptive they were to the information being provided.

Site 1, Mother, CF[-]: I's just down to the individual parent, really, because some will want to know everything and some, you know, for me, I couldn't really speak much and I needed to time to, like, digest the information so having the hospital appointment the next day was just right for us because it gave us a chance to think but not long term.

In terms of formatting, similar to the laboratory proformas, clinicians felt it would be useful if the checklists were on one side of A4.

Clinician 1, Site 2: If it was all on one page, I don't know, it may prompt me to do it more. I think when you're in the flow of conversation, you don't really read, do you? You just kind of know what it says and then I don't really turn the page. So, one page would be great.

Once implemented, clinicians also made various recommendations for things that could be added or removed from the communication checklists.

### Email/letter for parents
#### Coherence of the intervention
Most clinicians indicated that the email/letter for parents following the initial communication of the positive NBS result, had been useful to clarify the next steps for parents.

Clinician 2, Site 1: I like the emails, I do think they save time and they're structured professionally—I like the fact that it's in stages, like, 'This is what we've found and this is your appointment, what happens next?' and then the links…it just adds a nice touch to it.

Despite this, it was infrequently used during the implementation. This was possibly due to lack of clarity regarding who was responsible for sending the information to the family. Therefore, staff felt it would be important to clarify who was responsible for completing and disseminating the email/letter after the initial communication of the positive NBS result and regularly checking the links contained within still worked.

Clinician 4, Site 3: I think just so we all know what our roles are, so we know what responsibility that we take, and then if, going back to the email thing, if the email is being sent, then yes, who would ask the email address?

However, when shared with parents who had experienced the other interventions, they felt it would have been beneficial.

Site 3, Mother, CF[+]: To have that initial link to the website or just to be even told on the phone, you know, would be great to help, you know, understand and to help pass on the information to my family as well because even my parents didn't know what it was, and they Googled it.

These parents also indicated that the additional information provided about the hospital visit would have also been beneficial.

Site 3, Father, CF[+]: It does add to the stress of it a little bit, if you've got to look it up. At least, I suppose, if you've got all the information there, then that's something less to worry about on the day.

This view was echoed by parents who had not experienced the interventions who suggested that including information about why they had been contacted, details of when, where and with whom their appointment would be with and what would happen during the appointment as well as condition specific and reliable information sources would have been beneficial.

Site 2, Mother, MCADD[-]: I'd think, 'Jesus, how am I supposed to do this? What are the logistics of getting my child there? Where is it? Where do I park? How can I park?' So, yes, definitely a sheet like, I think, that would be extremely helpful.

Furthermore, both parents who had and had not experienced the interventions indicated this could have reduced their initial anxiety by providing reliable information resources. Parents who had experienced the other interventions stated due to not having sufficient information after receiving their child's positive NBS result, they used other information sources such as Google even though they had been advised not to which had added to their distress.

Site 3, Mother, CF[+]: They didn't really explain to me over the phone what it was so obviously I Googled it, worst decision ever.

This was also expressed by parents who had not experienced any of the interventions. Again, this was attributed to a lack of information provision when the NBS results was communicated to them but also fear related to what was perceived as a warning about Google:

Site 2, Mother and Father, IVA[-]: They kind of just give us the name of it, and then said, 'Don't Google

it.' Which left us-, you obviously think, 'Oh, why not Google it? It's obviously really bad.'

This often led to increased anxiety as the information was either inaccurate or out of date. Therefore, both sets of parents welcomed the idea of being sent relevant and reliable hyperlinks after the initial communication of the positive NBS result.

### Compatibility and resource use
In terms of compatibility with existing resources, some clinicians indicated that the email/letter was similar to approaches they already had in place and so did not foresee any difficulties with implementing it into practice. Other felt it represented an improvement on current processes in use and could be time saving. However, clinicians were concerned about the email/letter following communication of the initial positive NBS result meeting GDPR requirements.

> Clinician 1, Site 1: NHS.net, is secure, but their email isn't secure. It's about sending confidential information. So, if we're emailing a parent…we have to send an initial email which says, 'You are consenting to sending confidential information through potentially an insecure email.' We have to get that consent first before we send anything over email.

Clinicians were also concerned about the accessibility of the interventions, particularly the letter/email for parents following communication of a positive NBS result in terms of those with language or literacy barriers.

However, most clinicians felt that parents would be able to access the links in the email /letter.

> Clinician 4, Site 1: I think most of the population have got a smartphone that they use to browse, so there's no reason why they can't click on the links. So, I would be surprised if they didn't have the ability to do that.

### DISCUSSION
Implementing the interventions in the two study sites served to highlight the differences between the efficacy and effectiveness of the co-designed interventions[30]; some of the suggestions made during the co-design phase, were not fulfilled when implemented in practice. The impact of COVID-19 also led to issues with sustained engagement with staff which also hampered the implementation phase. This has been explored in more detail overall and for each of the co-designed interventions.

### Intervention fidelity
Auditing completed proformas in the current study revealed that these were only partially implemented; only the first page of the double-sided document was completed and completion of the first page ranged from 58%–76%. Completion of the communication checklists ranged from 43%–80%. During the co-design phase,

staff indicated that more detailed information provision would be beneficial to improve communication practices. However, in practice, feedback suggested that there may have been some ambiguity related to who was responsible for completing certain sections and evidence of duplication. Both of these may have acted as barriers to completion and may have accounted for the lower completion rates for certain sections. In addition, limited sustained engagement with staff has been identified previously as a barrier to standardisation in healthcare.[31 32] The level of staff engagement during the implementation phase differed both within and across study sites which could have contributed to this variation in completion rates.

### Implementation of the laboratory proforma
Site 1 was served by one NBSL, Sites 2 and 3 were served by the other NBSL. Communication of positive NBS results starts in the NBSLs who make referrals to clinicians who then communicate with parents. Therefore, in essence, staff in the NBSLs led the implementation of the co-designed interventions. Staff in Site 2 acted as champions for the co-designed interventions and even expressed the desire to keep using them after the study ended. It is known that having a champion to advocate for the 'new way of doing things'[32] can lead to interventions being implemented more effectively and this was certainly evident in the implementation phase. This also highlighted the importance of leadership and the ability of perceived leaders to control implementation in terms of facilitating or hindering the process. The role of champions providing leadership in NBS strategies has been highlighted previously.[33] This emphasises the importance of involving key stakeholders such as organisational leads and potentially policy-makers in any future studies.

Staff in Site 1 stated they were not aware that different referral forms were being used by NBSLs throughout the country and therefore the need for the intervention. Evidence suggests that when staff are dismissive of the evidence, this can reinforce resistance to implementation efforts.[32] Furthermore, convincing staff that there is a problem, and that the proposed solution is appropriate has been previously recognised as a barrier to healthcare improvement.[34] This was further hampered by clinicians in this site indicating that certain members of the NBSL had instructed others not to use the co-designed interventions. This was despite the fact that all clinicians who participated in implementation had been involved in the project from the outset and the perception therefore being that they had 'bought in' to the aims and objectives of the project. However, during implementation, these same clinicians acted as gatekeepers as the communication process started in the NBSL and if this section was not completed, it made it more difficult and potentially time consuming for the rest of the interventions to be completed. Leadership plays a vital role in successful implementation of complex interventions and respected individuals can play a vital role in encouraging colleagues across different professions.[32 34] Therefore, not having

buy in from what was considered top of the communication chain for positive NBS results, had the potential to hamper the implementation process.

The other sites were aware that different processes were being used nationally and felt that standardisation would be important for many reasons including: ease of completion; standardisation of communication; and transferability between NBSL. Potential problems included compatibility with existing computer systems; one site felt it would be preferable if the proforma could be automatically populated as this would be time saving and reduce the potential for errors. Other studies which have also attempted to standardise process in healthcare settings have highlighted similar issues. One study which attempted to implement a standardised policy for labelling of invasive tubing and lines in a UK region found that despite being seen as a common sense approach at the outset, numerous practical, social and cultural challenges hampered implementation.[31] Similar to the current study, some staff remained unconvinced of the need for the change. Furthermore, practical issues which challenged pre-existing norms, practices and procedures were also found to be a barrier to successful implementation.

### Communication of the positive NBS result: communication checklists and the email/letter template

Parents in the present study who had and had not experienced the co-designed checklist indicated that standardising communication of positive NBS results would be preferable to ensure consistency. Some clinicians also indicated that standardisation of initial communication with parents following a positive NBS result would be beneficial. Others felt that standardisation was not always possible or desirable as the communication was too nuanced and complex. As mentioned previously, studies have indicated that parental information needs regarding the NBS result were variable, condition specific[7] and individualised.[15] However, other studies have indicated that variability in the content and the method used for communication can lead to increased parental anxiety and distress.[4 6]

Clinicians who chose to not use the communication checklist stated that they had developed their own way of doing things and therefore did not see the purpose of it. This reiterates the impact of convincing staff that there is a problem that needs to be addressed but also that the proposed solution is appropriate.[34] In addition, an organisational culture comprising of staff resistant to trial new innovations has been cited as a potential barrier to successful implementation of new interventions.[32] This could have explained the clinicians' reticence to trial the checklist although this was not evident in other clinicians within the same team which perhaps suggests it may be more personal than organisational.

The objective of the communication checklist was to focus on the initial communication of the positive NBS result. However, during the co-design phase, clinicians and parents chose to develop checklists that included the first clinic visit as well as subsequent visits that included information about the NBS result, to facilitate communication between members of the multidisciplinary team (MDT). Despite this, when implemented, most staff only completed the checklists for the initial communication of the positive NBS result; when audited, completion of the initial checklist ranged from 43%–80%. Previous reviews of the literature have indicated that checklists can facilitate communication between MDT members in other settings such as cancer[35] and intensive care settings.[36] Both sets of parents, that is, those who had and had not experienced the co-designed checklist also indicated that checklists that covered ongoing communication regarding their child's positive NBS result could facilitate pacing and tailoring of information. Quantity and quality of the initial communication of the positive NBS result has been deemed problematic in previous studies[4 12 15] and suggests that guidance that meets parents needs but is also flexible may be preferable.

Regardless of the approach used, the skills and attributes of the person communicating the result was an important factor in terms of the communication of positive NBS results for all parents in the present study regardless of whether they had experienced the interventions or not. This has been highlighted in previous studies[4 11 37] and demonstrates the value of the interpersonal skills of the person communicating the positive NBS results. One clinician in the current study stated that these skills cannot be captured in any form of guidance or checklist. However, other staff felt that the communication checklists would be useful training aids for less experienced staff. However, it is acknowledged that while the checklists can act as a guide, and standardise and facilitate communication strategies, they cannot teach someone how to be empathetic to parental cues in what is undoubtedly a highly emotive encounter.

Almost all parents who had received a positive NBS result consulted the internet for information about their child's suspected condition even though they had been advised not to do so. This is similar to the findings of previous studies.[6 15] In order to remedy this, an email/letter outlining next steps and appropriate information sources was developed. Parents and staff indicated this would be useful to deter parents from accessing sites which were outdated and/or inaccurate as well as providing information they could share with family and friends. However, it was acknowledged that in order for this to continue to be successful, it would be vital to clarify who would be responsible for checking and updating these.

### Strengths and limitations

The current study has numerous strengths. This is the first known study that has used co-designed interventions to improve communication of positive newborn blood-spot screening results. HCPs involved in the present study were employed in three different NHS organisations, increasing transferability of the findings. Also, the study

included: HCPs involved in managing all nine conditions currently included in the NBS programme in England; and parents of children diagnosed with one of the nine conditions currently included in the NBS programme in England; previous work has mainly focused on CF and SCD. Members of the PPI advisory group and relevant charities contributed to the study design, data collection and analysis.

In terms of limitations, HCPs were recruited via email; those with a pre-existing interest in this topic may have been more likely to self-select into the study. These people may communicate results differently than providers who did not participate in the study. The researchers are experienced in this field which may have biased data collection and analysis. COVID-19 hindered implementation of the co-designed interventions and related data collection.

### Recommendations for future research and practice

Given the variability in terms of intervention fidelity across individuals and sites, further feasibility testing with additional sites, staff members and parents to inform and refine the interventions prior to a national evaluation would be beneficial.

However, the preliminary findings suggest the use of standardised, condition specific laboratory proformas and checklists for communicating positive NBS results to families would ensure that vital information is consistently relayed between HCPs and to families thereby reducing unnecessary variation. While most study participants thought the proformas and checklists were a positive improvement to current practice, this was not unanimous and several participants highlighted variation in practice and potential barriers with implementation. These need to be better understood ahead of implementation.

Guidance regarding reliable sources of further information for parents would also reduce alarm that can be caused by accessing unhelpful content on the internet immediately after the initial communication of the positive NBS result. This might include the use of specifically designed applications or other forms of 'easy to access' and helpful online information for parents. Keeping this information up to date can be a laborious task, and if done locally this could result in different quality of information being shared with parents; one of the main issues that the guidance would be trying to avoid.

### CONCLUSION

Unjustified variations in communication practices for positive NBS screening results continue to exist; these have the potential to cause real and repeated harms. Implementation of the co-designed interventions demonstrated that they have the potential to standardised communication of positive NBS result from NBSLs to clinical teams and then from clinicians to parents; this could improve parents' experience of receiving a positive NBS result. Implementation highlighted organisational

and contextual barriers to effective adoption of the co-designed interventions in practice.

**Author affiliations**
[1]School of Health Sciences, City University of London, London, UK
[2]Centre for Arts, Memory and Communities, Coventry University, Coventry, UK
[3]Health Services and Population Research, King's College London, London, UK
[4]Florence Nightingale Faculty of Nursing, Midwifery and Palliative Care, King's College, London, UK
[5]Paediatrics, University of Liverpool, Liverpool, UK
[6]Primary Care Unit, University of Cambridge, Cambridge, UK
[7]Department of Public Health and Primary Care, University of Cambridge, Cambridge, UK
[8]Division of Psychology and Mental Health, The University of Manchester, Manchester, UK
[9]Paediatric Psychology and Play Services, Great Ormond Street Hospital For Children NHS Foundation Trust, London, UK
[10]Pharmacy, Diagnostics and Genetics, Sheffield Children's Hospital NHS Foundation Trust, Sheffield, UK

**Acknowledgements** We would like to thank the Newborn Screening Laboratory Directors in England for agreeing to act as local principle investigators for this study as well as members of clinical teams who gave their valuable time. We would like to particularly acknowledge the Department of Clinical Chemistry and Newborn Screening at Sheffield Children's NHS Foundation Trust for allowing us to use their laboratory template which formed the basis of the laboratory proforma. We would like to acknowledge the CF teams at Sheffield Children's Hospital and King's College Hospital for sharing their templates and the suggested content offered by the Lead IMD & Newborn Screening Nurse at Birmingham Children's Hospital which has collectively helped with development of the communication checklists. We would also like to especially thank Gemma Hack and Tanya Gill, Paediatric Metabolic Clinical Nurse Specialists at St Thomas Hospital for their help with the email/letter template. We would also like to thank all the parents in the Public and Patient Involvement Advisory Group, for their invaluable input and particularly Celia Charlwood for Chairing this group. We would also like to thank the charities including; The Cystic Fibrosis Trust, the Sickle Cell Society, the National Society for PKU, The British Thyroid Foundation, Metabolic Support UK as well as members of the National Newborn Screening Programme and the NHS Sickle Cell Disease and Thalassaemia Screening Programme for their ongoing support during this work.

**Contributors** JC made substantial contributions to the conception and design of the work. She acquired and interpreted the data for the work. She was involved in drafting the work, approved the final version to be published and agrees to be accountable for all aspects of the work in ensuring that questions related to the accuracy or integrity of any part of the work are appropriately investigated and resolved. PH acquired and interpreted the data for the work. She was involved in drafting the work, approved the final version to be published and agrees to be accountable for all aspects of the work in ensuring that questions related to the accuracy or integrity of any part of the work are appropriately investigated and resolved. EO was involved interpreting the data for the work and revising the work critically for important intellectual content, approved the final version to be published and agrees to be accountable for all aspects of the work in ensuring that questions related to the accuracy or integrity of any part of the work are appropriately investigated and resolved. JRB, AS, SM, FF, LM, FU, MB and KS made substantial contributions to the conception and design of the work. They were involved in drafting the work, approved the final version to be published and agree to be accountable for all aspects of the work in ensuring that questions related to the accuracy or integrity of any part of the work are appropriately investigated and resolved.

**Funding** This study was supported by the National Institute for Health Research (NIHR) (Health Services and Delivery Research (project reference 16/52/25)).

**Disclaimer** The views expressed are those of the authors and not necessarily those of the NIHR or the Department of Health and Social Care.

**Competing interests** None declared.

**Patient consent for publication** Not required.

**Ethics approval** This study was approved by the London Stanmore ethics committee, reference 17/LO/2102.

**Provenance and peer review** Not commissioned; externally peer reviewed.

**Data availability statement** Data are available upon reasonable request. Data are available upon reasonable request from the corresponding author subject to restrictions to preserve anonymity and personal privacy [JC]. These data are not publicly available as they contain information that could compromise research participant privacy/consent. Data will be available beginning 1 year and ending 5 years after publication to researchers who propose a methodologically sound proposal. Proposals should be directed to j.chudleigh@city.ac.uk. To gain access, data requesters will need to sign a data access agreement.

**ORCID iDs**
Jane Chudleigh http://orcid.org/0000-0002-7334-8708
Stephen Morris http://orcid.org/0000-0002-5828-3563

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
