## [Reviewer comments · BMJ Open]

ARTICLE DETAILS

TITLE (PROVISIONAL)	A process evaluation of co-designed interventions to improve communication of positive newborn bloodspot screening results.
AUTHORS	Chudleigh, Jane; Holder, Pru; Moody, Louise; Simpson, Alan; Southern, Kevin; Morris, Stephen; Fusco, Francesco; Ulph, Fiona; Bryon, Mandy; Bonham, Jim; Olander, Ellinor

VERSION 1 – REVIEW

REVIEWER	Farr, Michelle University of Bristol, NIHR CLAHRC West, Bristol Medical School
REVIEW RETURNED	12-May-2021

GENERAL COMMENTS	This is a well written, detailed and thorough analysis of an interesting study. The introduction is clear and covers all necessary details. The methods are well set out and explained. The data analysis section is thorough and detailed. I wonder if at some points if it is overly long – can the professionals quotes be summarised and edited down at all, for the convenience of the reader? I thought that the patients quotes are really valuable and really helpful to have interviewed both people who have experienced and not experienced the intervention. The patients quotes really help to illustrate why this intervention is needed in spite of implementation difficulties. The study is well structured and framed with the use of NPT, Figure 1 and Interview questions clearly illustrating how NPT has been used to frame the study. Just one minor point may help with clarity: Inclusion and exclusion criteria: I thought it was interesting that: “HCPs (NBSL staff and clinicians) who had not been involved in NBS in the last 6 months or who had personal experience of receiving a positive NBS result were excluded.” – so this suggests that there may have been HCPs who have also had experience of receiving a positive NBS result but that these were excluded? Why were they excluded as it may have been interesting to explore their dual experiences of being a HCP and receiving a positive NBS result? Or have I misunderstood this sentence? Overall I commend the authors on such a comprehensive, detailed and clear manuscript. Well done!
--

REVIEWER	Potter, Beth University of Ottawa, Epidemiology and Community Medicine
REVIEW RETURNED	25-May-2021

GENERAL COMMENTS	This study aimed to implement and complete a process evaluation of three co-designed interventions to improve communication of positive newborn bloodspot screening results. These interventions included a laboratory pro forma, communication checklists, and
---

	email templates with informational resources. This was a careful study with interesting findings and the paper is well-written. The process evaluation was grounded in Normalisation Process Theory and a set of success criteria to guide the interview questions. The data were analyzed using framework analysis (staff) and thematic analysis (parents) in a robust process. I have a few comments and questions for the authors' consideration.  1. The authors note that the parents selected for the study were identified by the health care providers who had used the co-interventions. Did these health care providers identify a list of all possible parent respondents during the study window? If not, (i.e., if the parents were selectively identified), could this have implications for interpreting the findings? 2. Several study participants made suggestions for modifications to the interventions (e.g., having all of the information on one page, clarifying who should contribute the various pieces of information, who should be responsible for aspects such as sending out the email). Have these modifications been reviewed by the study team and PPI advisory group? Have any changes been made to the interventions as a result? What are the next steps for the interventions? 3. The level of staff engagement in the intervention implementation was quite variable and the authors also acknowledge that those willing to be interviewed may have been more engaged than others. I appreciated the thoughtful discussion of these points. One aspect that I think could be further explained is the role of the laboratory staff members who were interviewees with respect to leadership and ability to control implementation. Did these participants feel they had the ability to implement the checklists or were there 'gatekeepers' (also mentioned in the study I realize) who could either facilitate or hinder their efforts? Would it be worth (in future) discussing the interventions with organizational leads or even policy-makers at higher levels? 4. Were the results variable enough across individuals and sites that the authors would recommend further process evaluation with additional sites, staff members, and parents to inform a refining of the interventions? 5. Equity arose as an issue in some different ways across the interviews. On the one hand, a parent participant commented that a strength of a standardized intervention is that it facilitates access to the same information regardless of religion, education, etc. On the other hand, a clinician worried that with the email intervention in particular, there is an assumption about English literacy. Was accessibility a factor that was considered in the intervention co-design phase?
--	---

REVIEWER	Nicholls, Stuart University of Ottawa, Epidemiology and Community Medicine
REVIEW RETURNED	02-Jun-2021

GENERAL COMMENTS	Manuscript ID bmjopen-2021-050773A process evaluation of co-designed interventions to improve communication of positive newborn bloodspot screening result
--

In the present study the authors took a multifaceted approach to evaluating three interventions (standardised laboratory proformas, communication checklists, and an email/letter template) that were designed to improve the communication of a positive newborn screen. The manuscript is quite dense and requires an amount of back and forth given the varied components and limited details about the interventions and their development. Moreover, despite the qualitative analysis there appeared to be limited thematic results made explicit – pulling out the key themes of challenges and strengths of the implementation, with appropriate headings or subheadings, would certainly help orient the reader. Finally, I was expecting more critical reflection on the implementation and specific assessment of the challenges and potential remedies. However, this really doesn't come through and an implementation science lens could help here

Below are some specific comments that I hope are useful to the authors.

ABSTRACT

The abstract indicates 20 parents were interviewed, but the text indicates that while 2 were invited only 18 were interviewed. Similarly, the abstract indicates 31 HCPs were interviewed, but the text says 24.

INTRODUCTION

The authors claim that “However, to date, no studies have focussed on designing or exploring strategies that can improve this process [of imparting a result] (page 6 of 49, lines 5-7). However several studies examining communication do exist. For example the following studies have specifically address the process of communicating a positive result:

Davis TC, Humiston SG, Arnold CL, et al. Recommendations for effective newborn screening communication: results of focus groups with parents, providers, and experts. *Pediatrics* 2006; 117: S326-340. DOI: 10.1542/peds.2005-2633M.

Collins JL, La Pean A, O'Tool F, et al. Factors that influence parents' experiences with results disclosure after newborn screening identifies genetic carrier status for cystic fibrosis or sickle cell hemoglobinopathy. *Patient Educ Couns* 2013; 90: 378-385. DOI: 10.1016/j.pec.2011.12.007.

Salm N, Yetter E, Tluczek A. Informing parents about positive newborn screen results: Parents' recommendations. *Journal of Child Health Care* 2012; Vol 16, Issue 4

METHODS

While the authors point the readers to the previous work, it would be helpful to have some description of the implementation process. Was there an underlying theory to the interventions, or the way in which the interventions were delivered? Was training given or other supports for the implementation process – or was it simply the provision of the documents?

What was the rationale for including parents who had not experienced the intervention? Given the goal was to implement and evaluate an implementation, it isn't clear how these parents

fulfil that aim if they cannot provide feedback on the experience, unless there is a comparative analysis between those who did and did not experience the use of the intervention, but this was not conducted. More description of this rationale would be helpful.

DATA ANALYSIS

Regarding the analysis of the proforma, how were these selected? Did they send all completed forms, or a sample? If the labs were able to select the examples surely this is a major source of bias? They could simply pick the most complete versions? Please can you say more about why this approach was taken as well as how more details about the details of the fidelity assessment; it would also be helpful to know the proportion of forms received compared to the number actually completed or the number of positive cases. This would provided better context of whether this was a highly selected number of forms or a more complete assessment

Page 7-8, the authors state that “Following this, the same two members of the research team coded the remainder of eth transcripts using the code book.” – can the authors clarify if the remaining transcripts were divided among the two reviewers, or if the transcripts were all dual coded? How were changes to the codebook communicated – was there consensus throughout? E.g. after a certain number of transcripts?

RESULTS

Regarding the fact that teams in Site 1 and Site 3 did not implement the co-designed interventions (assuming this is the checklist or the template), it would be helpful to have a better description of how the follow up works; there is no mention of teams or structure to inform the reader, and there are interviewees from each site. As such, it isn't clear who did or did not respond – were the treating different conditions etc? More information would be helpful here.

In terms of Table 3, the authors note that they only include the initial communication checklist in the assessment, but the mere fact that the other checklists were not used is clearly relevant to the assessment of the intervention fidelity

Page 13 seems to start with a quote about the laboratory proformas (quote from NBSL, Site 2, participant 1) – is that correct? This seems to be mixed in with discussion of the communication checklist.

For the average completion rates, please can you present the range of completion also – e.g. for the 7 CF form present not only the average completion, but also the range. This again would help to have a better understanding of fidelity.

It would also be more helpful to understand if there were specific items that were consistently not completed. This is hinted at in some interviews (where form items are noted as being redundant). Again, thinking about recommendations and revisions it would seem that this data would be very important.

While feedback from the quotes about the laboratory proforma suggests a positive disposition, there is alack of discussion about the implementation and what constituted full or partial implementation (which would seem relevant to any assessment).

Moreover, how do the authors square the circle of the positive disposition but the fact that most forms had roughly two thirds completion of side 1 and were almost all incomplete for side 2?

On that note, could the authors examine whether fidelity improved over time as familiarity increased (e.g. to look at whether the forms had increased completeness as time went on)?

On page 16 the text states “Responses from parents who had experienced interventions are denoted by (+), and parents who had not experienced interventions by (-)” – there is no context for this sentence and it seems like this should be more of a follow on to a broader introduction to the section.

The section Purpose of the Intervention, seems overly long and the relevance isn't clear- in many places it seems like something that should have been done prior to the development, it is exploring (at least in the case of parents who had not experienced the intervention) the potential utility or perspectives and I am unclear as to how this informs a process evaluation.

In fact, a challenge with the results is that while they are sorted by the different interventions, they don't really help to orient the reader to the challenges identified or key outcomes of the evaluation. For example, there is no thematic groupings despite the thematic analysis. Consequently, one is left wondering what the themes were that were identified in the analysis (the results are simply presented as the intervention component broken down by views on the purpose and then compatibility and resource use.

DISCUSSION

The discussion is similarly lacking in concrete elements of assessment or next steps. Indeed, despite the low levels of uptake and the missing data issues, the authors argue that the interventions could be useful without addressing the key challenges raised or explaining how it might be adapted to address the deficiencies identified. There is no real attempt to examine why the sites differed, or look at why some teams chose not to implement the checklists and whether there could be any explanatory factors there.

Further there is a lack of reflection on aspects of implementation science and the development of the intervention and its implementation. Rather it is a summary of findings with limited drawing on other studies. This weakens the manuscript as it really doesn't offer critical reflect or proposal of what can or should be done differently. For example, during the development the number of communication checklists were expanded: how was this decided? Were clinicians involved in that process? This sort of reflection would be useful.

Leadership is mentioned, but is not mentioned elsewhere – did it come up in the interviews, or is this speculation that there wasn't leadership of the intervention. Again, a lack of description about how the intervention was rolled out precludes a more detailed assessment. One would have thought gatekeeper buy in would have been established at the outset or during intervention development.

	The comment that clinicians noted NBSL staff instructed others not to use the interventions does not appear in the results. This reflects amore active attempt to undermine the intervention which is not drawn out. Similarly it is suggested that skills and attributes of the communicator are important. This is also not clearly communicated in the results.
--	--

VERSION 1 – AUTHOR RESPONSE

Reviewer: 1	
This is a well written, detailed and thorough analysis of an interesting study. The introduction is clear and covers all necessary details. The methods are well set out and explained. The data analysis section is thorough and detailed.	Thank you for your positive feedback
I wonder if at some points if it is overly long – can the professionals quotes be summarised and edited down at all, for the convenience of the reader?	Some of the HCP quotes have been removed to reduce the length (pages 12-24)
I thought that the patients’ quotes are really valuable and really helpful to have interviewed both people who have experienced and not experienced the intervention. The patients’ quotes really help to illustrate why this intervention is needed in spite of implementation difficulties.	Thank you for your positive feedback
The study is well structured and framed with the use of NPT, Figure 1 and Interview questions clearly illustrating how NPT has been used to frame the study.	Thank you for your positive feedback
Just one minor point may help with clarity: Inclusion and exclusion criteria: I thought it was interesting that: “HCPs (NBSL staff and clinicians) who had not been involved in NBS in the last 6 months or who had personal experience of receiving a positive NBS result were excluded.” – so this suggests that there may have been HCPs who have also had experience of receiving a positive NBS result but that these were excluded? Why were they excluded as it may have been interesting to explore their dual experiences of being a HCP and receiving a positive NBS result? Or have I misunderstood this sentence?	HCPs who had personal experience of receiving a positive NBS result were excluded. This was because we wanted to explore parental experiences and HCP experiences separately and did not want to muddy the two.
Overall I commend the authors on such a comprehensive, detailed and clear manuscript. Well done!	Thank you for your positive feedback

Reviewer 2:	
This study aimed to implement and complete a process evaluation of three co-designed interventions to improve communication of positive newborn bloodspot screening results. These interventions included a laboratory pro forma, communication checklists, and email templates with informational resources. This was a careful study with interesting findings and the paper is well-written. The process evaluation was grounded in Normalisation Process Theory and a set of success criteria to guide the interview questions. The data were analyzed using framework analysis (staff) and thematic analysis (parents) in a robust process.	Thank you for your positive feedback
The authors note that the parents selected for the study were identified by the health care providers who had used the co-interventions. Did these health care providers identify a list of all possible parent respondents during the study window? If not, (i.e., if the parents were selectively identified), could this have implications for interpreting the findings?	Apologies, this was not explained very well and has now been clarified as follows (page 6): “HCPs provided all parents who had experienced the interventions with an information sheet about the study and asked if they could share their details with the research team. Once the details were received, parents were contacted by the research team and invited to participate in the study. In addition, a purposeful sample of parents who had received a positive NBS result without the co-designed interventions were also identified by HCPs. Written informed consent was obtained from all participants.”
Several study participants made suggestions for modifications to the interventions (e.g., having all of the information on one page, clarifying who should contribute the various pieces of information, who should be responsible for aspects such as sending out the email). Have these modifications been reviewed by the study team and PPI advisory group? Have any changes been made to the interventions as a result? What are the next steps for the interventions?	We were really pleased with the additional feedback received by parents and HCPs when these were implemented in practice. Our future plans include further feasibility testing +/- a national study to further evaluate the interventions in additional sites with staff members, and parents to allow us to refine them (page 29). We are in the process of working with our PPI group to submit a further application. In terms of next steps, in the revised section titled “Recommendations for future research and practice”, we have suggested further feasibility

	testing and a future national evaluation study (page 29).
The level of staff engagement in the intervention implementation was quite variable and the authors also acknowledge that those willing to be interviewed may have been more engaged than others. I appreciated the thoughtful discussion of these points. One aspect that I think could be further explained is the role of the laboratory staff members who were interviewees with respect to leadership and ability to control implementation. Did these participants feel they had the ability to implement the checklists or were there 'gatekeepers' (also mentioned in the study I realize) who could either facilitate or hinder their efforts? Would it be worth (in future) discussing the interventions with organizational leads or even policy-makers at higher levels?	Thank you for your positive feedback. Thank you for your suggestions. In our future work this would be a really valuable recommendation to take forward. The laboratory staff in one study site were definitely viewed as 'gate keepers' who could either hinder or facilitate the implementation of the interventions. We have explored this further in the discussion as follows (page 25): "This also highlighted the importance of leadership and the ability of perceived leaders to control implementation in terms of facilitating or hindering the process. In addition, the importance of involving key stakeholders such as organisational leads and potentially policy-makers in any future studies."
Were the results variable enough across individuals and sites that the authors would recommend further process evaluation with additional sites, staff members, and parents to inform a refining of the interventions?	As per the comment above, we were really pleased with the additional feedback received by parents and HCPs when these were implemented in practice. Our future plans include further feasibility testing +/- a national evaluation study to further evaluate the interventions in additional sites with staff members, and parents to allow us to refine them. We have added the following to reflect this (page 29): "Given the variability in terms of intervention fidelity across individuals and sites, further feasibility testing with additional sites, staff members, and parents to inform and refine the interventions prior to a national evaluation would be beneficial."
Equity arose as an issue in some different ways across the interviews. On the one hand, a parent participant commented that a strength of a standardized intervention is that it facilitates access to the same information	Accessibility in terms of language and understanding was part of the success criteria (coherence) that was explored with HCPs. As mentioned, this was raised by HCPs as a potential issue but not highlighted by parents as

regardless of religion, education, etc. On the other hand, a clinician worried that with the email intervention in particular, there is an assumption about English literacy. Was accessibility a factor that was considered in the intervention co-design phase?	an actual issue in practice. This is something else that we believe would benefit from being explored further during further feasibility testing +/- a national evaluation study.
Reviewer 3:	
In the present study the authors took a multifaceted approach to evaluating three interventions (standardised laboratory proformas, communication checklists, and an email/letter template) that were designed to improve the communication of a positive newborn screen. The manuscript is quite dense and requires an amount of back and forth given the varied components and limited details about the interventions and their development. Moreover, despite the qualitative analysis there appeared to limited thematic results made explicit – pulling out the key themes of challenges and strengths of the implementation, with appropriate headings or subheadings, would certainly help orient the reader. Finally, I was expecting more critical reflection on the implementation and specific assessment of the challenges and potential remedies. However, this really doesn't come through and an implementation science lens could help here	Some of the HCP quotes have been removed as per the comments from Reviewer 1 (page 12-24). The key themes for the interventions were (page 13): Coherence of the intervention Compatibility and resource use Ease of completion and layout A sentence has been added at the beginning of the results section to make this clearer as follows (page 13): “Implementation of the interventions Three themes were identified in relation to the implementation of the laboratory proforma and the communication checklists: Coherence of the intervention; compatibility and resource use; and ease of completion and layout. The first two themes were also applicable in relation to implementation of the email/letter to parents.” Further critical reflection has been added to the discussion (pages 24-29). In addition, we have amended the final section so it now includes, “Recommendations for future research and practice” to consider future research as well as practice implications (page 29).

ABSTRACT: The abstract indicates 20 parents were interviewed, but the text indicates that while 20 were invited only 18 were interviewed. Similarly, the abstract indicates 31 HCPs were interviewed, but the text says 24.	Apologies, this has been corrected (page 2)
INTRODUCTION: The authors claim that “However, to date, no studies have focussed on designing or exploring strategies that can improve this process [of imparting a result] (page 6 of 49, lines 5-7). However several studies examining communication do exist. For example the following studies have specifically address the process of communicating a positive result: Davis TC, Humiston SG, Arnold CL, et al. Recommendations for effective newborn screening communication: results of focus groups with parents, providers, and experts. Pediatrics 2006; 117: S326-340. DOI: 10.1542/peds.2005-2633M. Collins JL, La Pean A, O'Tool F, et al. Factors that influence parents' experiences with results disclosure after newborn screening identifies genetic carrier status for cystic fibrosis or sickle cell hemoglobinopathy. Patient Educ Couns 2013; 90: 378-385. DOI: 10.1016/j.pec.2011.12.007. Salm N, Yetter E, Tluczek A. Informing parents about positive newborn screen results: Parents' recommendations. Journal of Child Health Care 2012; Vol 16, Issue 4	Thank you for highlighting this. This has been clarified as follows to illustrate that while studies have explored experiences and preferences, none have subsequently implemented or evaluated any interventions in practice as a result (page 5). “However, to date, while studies have explored experiences of receiving positive NBS result and strategies for improving communication, no studies have focussed on designing implementing or evaluating such strategies in practice.”
METHODS: While the authors point the readers to the previous work, it would be helpful to have some description of the implementation process. Was there an underlying theory to the interventions, or the way in which the interventions were delivered?	The interventions were developed using Experience-based co-design. This has been added to the methods as follows (page 5): “Experience-based co-design was used to develop co-designed interventions to improve communication of positive NBS result to families.”

Was training given or other supports for the implementation process – or was it simply the provision of the documents?	In addition, further information about associated training has been added to the methods and results sections as follows: Methods (page 5): “Staff in each study site were able to choose from a variety of training options including face-to-face (in person or remotely) individual or group training (this was condition specific to ensure the correct interventions were presented to the relevant staff) via narrated PowerPoint presentations and/or annotated PowerPoint presentations. These training materials were also made available on the study blog.” Results (page 9): “Training Training for the implementation of the co-designed intervention was undertaken with 23 staff in study site 1, nine in study site 2, and 14 staff in study site 3. Forty-one staff were trained during face to face sessions and five were trained online. Staff were asked to provide feedback at the end of the training sessions using a five-point Likert scale consisting of statements ranked from strongly agree to strongly disagree. Twenty-nine staff (63%) provided feedback. These were scored numerically so that strongly agree was awarded a score of 5 and strongly disagree was awarded a score of 1; a score of 3 would therefore have indicated a neutral response, over 3 would indicate positive feedback and under 3, negative feedback. Responses ranged from 4.3 to 4.6 (median 4.5). “
METHODS: What was the rationale for including parents who had not experienced the intervention? Given the goal was to implement and evaluate an implementation, it isn't clear how these parents fulfil that aim if they cannot provide feedback on the experience, unless there is a comparative analysis between those who did and did not experience the use of the intervention, but this was not conducted. More description of this rationale would be helpful.	The rationale was to see if there were any differences between parents who had and had not experienced the co-designed interventions. However, we had not made this explicit; this has now been included (page 6, 26, 27).
DATA ANALYSIS: Regarding the analysis of the proforma, how were these selected? Did they send all completed forms, or a sample? If the labs were able to select the examples surely this is a major source of bias? They could simply pick the most complete	HCPs were asked to send all completed forms to the research team. This has been changed in the text as follows (page 7): “Staff were asked to send the research team all completed laboratory proformas and

versions? Please can you say more about why this approach was taken as well as how more details about the details of the fidelity assessment; it would also be helpful to know the proportion of forms received compared to the number actually completed or the number of positive cases. This would provided better context of whether this was a highly selected number of forms or a more complete assessment	communication checklists so these could be audited in terms of accuracy and completeness. These were redacted to maintain patient confidentiality.” .”
DATA ANALYSIS: Page 7-8, the authors state that “Following this, the same two members of the research team coded the remainder of eth transcripts using the code book.” – can the authors clarify if the remaining transcripts were divided among the two reviewers, or if the transcripts were all dual coded? How were changes to the codebook communicated – was there consensus throughout? E.g. after a certain number of transcripts?	Further information on this process has been provided as follows (page 8): “Following this, the remainder of the transcripts were divided between the same two members of the research team for coding. This was an ongoing, iterative process; new codes were developed and the definition of codes refined as analysis progressed. Any proposed changes to the code book were discussed where relevant after each interview transcript had been analysed to ensure consensus; the final version of the code book (version 3) was developed after a further three transcripts had been analysed. Once coding had been completed, all data for each code were compared to ensure consistency in coding.”
RESULTS: Regarding the fact that teams in Site 1 and Site 3 did not implement the co-designed interventions (assuming this is the checklist or the template), it would be helpful to have a better description of how the follow up works; there is no mention of teams or structure to inform the reader, and there are interviewees from each site. As such, it isn’t clear who did or did not respond – were the treating different conditions etc? More information would be helpful here.	Thank you for your suggestion. However, we are reluctant to provide further details of the teams and structure as we believe this could make the study sites and even possibly members of laboratories and clinical teams identifiable which would breach our ethical approval.
RESULTS: In terms of Table 3, the authors note that they only include the initial communication checklist in the assessment, but the mere fact that the other checklists were not used is clearly relevant to the assessment of the intervention fidelity	The following has been added to make this more explicit (page 12): “However, the second page, which contained further information about the NBS test results and a checklist regarding completion of the referral process was not completed indicting a lack of fidelity for this section of the intervention.” In addition, the following sentence has been added to the results (page 12): “Therefore, while in theory staff favoured the idea of having more information in one place about the child and family, this was not borne out in practice.”

RESULTS: Page 13 seems to start with a quote about the laboratory proformas (quote from NBSL, Site 2, participant 1) – is that correct? This seems to be mixed in with discussion of the communication checklist.	This is correct, it relates to the comments above about the laboratory staff feeling the proforma should be on one side of A4 and therefore offering an explanation for why the second side had not been completed.
RESULTS: For the average completion rates, please can you present the range of completion also – e.g. for the 7 CF form present not only the average completion, but also the range. This again would help to have a better understanding of fidelity.	These have been added (page 11).
RESULTS: It would also be more helpful to understand if there were specific items that were consistently not completed. This is hinted at in some interviews (where form items are noted as being redundant). Again, thinking about recommendations and revisions it would seem that this data would be very important.	The following has been added (page 12): “The most common items not completed on the laboratory proforma included: the case/laboratory ID; the designation of the person making the referral; legacy names (CF); HbS mutations (SCD); GP phone number; results of other conditions included in NBS; and feedback from the clinical team.” AND “The most common items not completed on the initial communication checklist included: diagnosis date/age; weight; gene mutations (SCD); GP details; health visitor details; summary information from the first contact.”
RESULTS: While feedback from the quotes about the laboratory proforma suggests a positive disposition, there is a lack of discussion about the implementation and what constituted full or partial implementation (which would seem relevant to any assessment). Moreover, how do the authors square the circle of the positive disposition but the fact that most forms had roughly two thirds completion of side 1 and were almost all incomplete for side 2?	The following has been added to address this discrepancy between how the proforma was perceived in theory and how it was used in practice (page 25): “During the co-design phase, staff indicated that more detailed information provision would be beneficial to improve communication practices. However, in practice, feedback suggested that there may have been some ambiguity related to who was responsible for completing certain sections and evidence of duplication. Both of these may have acted as barriers to completion and may have accounted for the lower completion rates for certain sections.
RESULTS: On that note, could the authors examine whether fidelity improved over time as familiarity increased (e.g. to look at whether the forms had increased completeness as time went on)?	It is not evident that fidelity increased over time; causes for differences in completion are not clear even though the majority of the proformas for instance were completed in one study site (as they chose to continue to use them from March to December i.e. during the first lockdown due to the pandemic). However, due to C-19, they were only implemented for 4 months (with a six month interval in the middle) in the other sites which also makes this difficult to examine in depth. This further supports why further feasibility testing +/- a future national study

	would be beneficial; to explore in greater detail why certain parts of the proforma were completed and others not and why variation existed.
RESULTS: On page 16 the text states “Responses from parents who had experienced interventions are denoted by (+), and parents who had not experienced interventions by (-)” – there is no context for this sentence and it seems like this should be more of a follow on to a broader introduction to the section.	This has been moved to the section about ‘Interviews with parents’. Thank you for pointing this out (page 10).
RESULTS: The section Purpose of the Intervention, seems overly long and the relevance isn’t clear- in many places it seems like something that should have been done prior to the development, it is exploring (at least in the case of parents who had not experienced the intervention) the potential utility or perspectives and I am unclear as to how this informs a process evaluation.	Some of the HCP quotes have been removed to reduce the length of the manuscript as suggested by Reviewer 1 (pages 12-24). This was one of the themes identified from the data in relation to how clinicians and parents perceived the interventions when used in practice. In hindsight, ‘Coherence of the intervention’ may have been a better title for this theme as this is more in keeping with Normalisation Process Theory. This has been altered accordingly (page 13).
RESULTS: In fact, a challenge with the results is that while they are sorted by the different interventions, they don’t really help to orient the reader to the challenges identified or key outcomes of the evaluation. For example, there is no thematic groupings despite the thematic analysis. Consequently, one is left wondering what the themes were that were identified in the analysis (the results are simply presented as the intervention component broken down by views on the purpose and then compatibility and resource use.	The key themes for the interventions were: Coherence of the intervention Compatibility and resource use Ease of completion and layout A sentence has been added at the beginning of the results section to make this clearer as follows (page 13): “Implementation of the interventions Three themes were identified in relation to the implementation of the laboratory proforma and the communication checklists: Coherence of the intervention; compatibility and resource use; and ease of completion and layout. The first two themes were also applicable in relation to implementation of the email/letter to parents.”
DISCUSSION: The discussion is similarly lacking in concrete elements of assessment or next steps. Indeed, despite the low levels of uptake and the missing data issues, the authors argue that the interventions could be useful without addressing the key challenges raised or explaining how it might be adapted to address the deficiencies identified.	We have added next steps into the amended section on “Recommendations for future research and practice.” (Page 29) This has been explored further in the discussion and in the recommendations where it is suggested further feasibility testing would enable us to better understand fidelity (Pages 24-29).

There is no real attempt to examine why the sites differed, or look at why some teams chose not to implement the checklists and whether there could be any explanatory factors there.	This has been explored further in the discussion (pages 24-29)
DISCUSSION: Further there is a lack of reflection on aspects of implementation science and the development of the intervention and its implementation. Rather it is a summary of findings with limited drawing on other studies. This weakens the manuscript as it really doesn't offer critical reflect or proposal of what can or should be done differently. For example, during the development the number of communication checklists were expanded: how was this decided? Were clinicians involved in that process? This sort of reflection would be useful.	This has been elaborated upon in the discussion (pages 24-29). This has been clarified in the results (page 12) and discussion (page 27).
DISCUSSION: Leadership is mentioned, but is not mentioned elsewhere – did it come up in the interviews, or is this speculation that there wasn't leadership of the intervention. Again, a lack of description about how the intervention was rolled out precludes a more detailed assessment. One would have thought gatekeeper buy in would have been established at the outset or during intervention development.	The following has been added to clarify this (page 26): "This was despite the fact that all clinicians who participated in implementation had been involved in the project from the outset and the perception therefore being that they had 'bought in' to the aims and objectives of the project. However, during implementation, these same clinicians acted as gatekeepers as the communication process started in the NBSL and if this section was not completed, it made it more difficult and potentially time consuming for the rest of the interventions to be completed."
DISCUSSION: The comment that clinicians noted NBSL staff instructed others not to use the interventions does not appear in the results. This reflects a more active attempt to undermine the intervention which is not drawn out.	The following has been added to the results, to make this more explicit (page 19): "This included staff instructing others not to use the checklists as they were seen as a 'tick box exercise'."
DISCUSSION: Similarly, it is suggested that skills and attributes of the communicator are important. This is also not clearly communicated in the results.	The following sentence has been added to the results section to make this more explicit (page 17): "...thus, highlighting the perceived importance of the skills and attributes of the person communicating the result."

VERSION 2 – REVIEW

REVIEWER	Farr, Michelle University of Bristol, NIHR CLAHRC West, Bristol Medical School
REVIEW RETURNED	28-Jul-2021

GENERAL COMMENTS	I consider this revised manuscript suitable for publication. The authors have thoroughly revised the article in line with all reviewers comments and it is ready for publication.
---

REVIEWER	Farr, Michelle University of Bristol, NIHR CLAHRC West, Bristol Medical School
REVIEW RETURNED	28-Jul-2021

GENERAL COMMENTS	I consider this revised manuscript suitable for publication. The authors have thoroughly revised the article in line with all reviewers comments and it is ready for publication.
---